# Semantic-guided Diffusion Prototypical Network for Few-shot Classification

Chuxin Zhang
*School of Computer Science and Engineering*
*Tianjin University of Technology*
Tianjin 300384, China
chuxinzhang@stud.tjut.edu.cn

Jing Li*
*School of Computer Science and Engineering*
*Tianjin University of Technology*
Tianjin 300384, China
jing.li.2003@gmail.com

*Abstract*—Few-shot learning recognizes unlabeled samples from new classes using only a few of samples. Many recently proposed approaches have made progress based on meta-learning. However, current methods often overlook the category information within the query set and thus the obtained prototype is always unreliable. To tackle this issue, we design a Semantic-guided Diffusion Prototypical Network (SDPN) to generate representative prototypes for few-shot classification. Specifically, we leverage self-supervised learning to pre-train the feature extractor, thus obtaining accurately visual features. Furthermore, we introduce a semantic-guided diffusion process that aims to generate semantic features of the query set from random noise for a new task. Then, we introduce a visual-semantic fusion strategy that involves the alignment of semantic features with visual features to obtain representative prototypes that correspond to each image. We perform comprehensive experiments on miniImageNet and tieredImageNet datasets, and the results demonstrate that SDPN achieves enhancements in comparison to state-of-the-art methods.

*Index Terms*—Few-shot Learning, Diffusion Model, Meta-learning, Prototypical Network, Multi-modal

## I. INTRODUCTION

Deep learning techniques typically require substantial annotated data for training a model in order to achieve satisfactory performance for different computer vision tasks. Nevertheless, in practical applications, collecting such an extensive variety of labeled datasets is impractical and costly. Consequently, few-shot learning (FSL), which adapts to novel classes by training a model using a group of base classes with limited labeled data, has become an important research task. It is similar to the nature of human learning, where individuals can rapidly learn novel concepts based on limited samples [1].

Generally, FSL methods can be classified into three types: metric-based, meta-learning-based, and semantic-based methods. These methods all use the episode mechanism in the few-shot classification (FSC) task. The objective of FSC is to categorize unlabeled samples (the query set) using limited labeled samples (the support set). Metric-based methods [2], [3], [4] learn a task-agnostic embedding that can be used to calculate the distance between samples. Meta-learning methods [5], [6] handle the FSL problem with a two-step

This work was supported by National Natural Science Foundation of China under Grant 62373280.

*Corresponding author

learning process that involves meta-training and meta-testing. Specifically, the learning model will be trained with numerous independent supervised FSC tasks to learn how to adapt to novel tasks during the meta-training stage. Afterwards, the model is tested on a new unseen task during the meta-testing stage. Semantic-based methods [7] aim to extract meaningful features from the label information and combine visual and semantic features into a representative prototype with limited samples. To improve the accuracy of classification, some methods pre-train feature extractors on base classes to extract discriminative feature representations. Chen et al. [8] introduced an additional pre-training step to greatly enhance the accuracy of FSC tasks. This indicates that applying pre-training is critical for obtaining representative prototypes for FSC tasks.

Substantial advancements have been achieved in FSC tasks by minimizing the cross-entropy loss of base class labels in the pre-training and meta-learning stages. However, the optimized model can only solve the FSC tasks of base class, but cannot generalize well to novel classes. Furthermore, the majority of current approaches rely on a single modality, which ignores the benefits of combining multiple modalities, such as visual data and semantic information, for enhancing the performance of FSC. Recently, some methods [9], [10] have been proposed to generate more representative prototypes by combining the support set's visual and semantic features. Nevertheless, they did not consider the relationship between the semantic and visual modality of the query sample since its label is not available in the FSC tasks. In fact, the human perceptual system possesses a distinctive mechanism for visual perception, which can identify semantic information of new categories and promote the learning of new categories using prior knowledge. Therefore, it is necessary to design a new model that can effectively combine and utilize semantic and visual features of the query set to improve prototype representation.

This paper introduces the Semantic-guided Diffusion Prototypical Network (SDPN) for few-shot classification. The SDPN employs a semantic-guided diffusion process to generate semantic features of the query set. Then, it combines these semantic features with visual features to create more accurate prototypes for few-shot classification. There are three stages that make up the proposed SDPN, which include pre-training,

forward diffusion, and reverse diffusion. The feature extractor is trained on base classes via self-supervised learning during the pre-training stage, which can help extract more representative features. In the forward diffusion stage, for the support set, we utilize CLIP [11] to extract the features of class labels as semantic features, and obtain the images' visual features from the pre-trained feature extractor. We employ meta-learning on the support set to execute a generative process that smoothly transforms visual features into semantic features. In the reverse diffusion stage, we reconstruct the semantic features specific to the query set by iteratively denoising on a random noise. Finally, we fuse visual and semantic features of the support set and query set to generate accurate query prototypes and support prototypes based on the fusion strategy. We predict the label of the query sample by measuring the similarities between each query prototype and support prototype. The primary contributions of our paper are as follows:

- We use self-supervised learning to train the feature extractor in the pre-training stage, with the goal of acquiring high-quality visual features of samples.
- We propose a semantic-guided diffusion process that can generate semantic features of the query set by conditioning random noise on a limited sample of a given new task.
- We design a fusion strategy that combines the semantic and visual features to obtain representative prototypes of samples.
- We perform comprehensive experiments on two challenging benchmarks, miniImageNet [12] and tieredImageNet [13], to verify the performance of our framework.

## II. RELATED WORKS

### A. Few-shot Learning

Few-shot learning (FSL) aims to effectively adapt novel tasks using a few of labeled data. Representative FSL approaches include three types: metric-based, meta-learning-based and semantic-based methods. Specifically, Metric-based methods [2], [4], [12] learned a shared feature space, measuring the similarity between samples using a specific distance measure, such as Cosine similarity distance, Earth Mover distance and Euclidean distance. Snell et al. [2] introduced the prototypical network, a method that learns a metric space can be used to obtain a prototype of each class, computing the Euclidean distance between features of each category. The meta-learning-based methods [5], [6] aim to learn an initial parameter of the model, in which the learner samples a series of training tasks from the base class for training and further adapted to novel FSC tasks. Finn et al. [5] designed a framework known as MAML, which learns an initialization parameter of the base-learner in order to enable quicker adaptation to novel tasks. Semantic-based methods [9], [10] explore how to combine visual and semantic features for FSC tasks. They demonstrated that using semantic features, such as class names, can capture more accurate characteristics of the classes. Xing et al. [9] introduced a method that integrates

visual and textual modalities by calculating an adjustable mixture coefficient.

### B. Contrastive Language-Image Pre-training

The Contrastive Language-Image Pre-training (CLIP) [11] aims to provide the models with better semantic understanding and generalization capabilities through contrastive learning between language and image modalities. It has achieved remarkable results in image classification, visual question answering and semantic segmentation. In FSC tasks, the CLIP extracts semantic features of the support set, which helps improve the performance of FSL models. Gao et al. [14] designed the CLIP-Adapter, which enhances the frozen CLIP model for downstream FSC tasks by incorporating a multi-layer perceptron (MLP) and integrating language knowledge into the output. Zhang et al. [15] proposed Tip-Adapter, a nonparametric adaptive method of CLIP, which can extract knowledge from pre-trained CLIP with less data in FSC tasks. In our work, we utilize CLIP as the text encoder to extract semantic features from the samples without additional training parameters.

### C. Diffusion Model

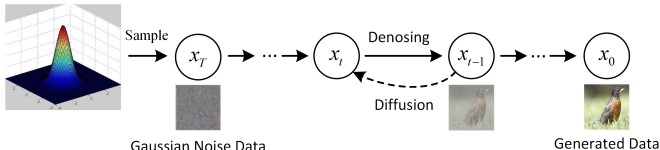

Fig. 1. The diffusion processes.

Diffusion models [16], [17] generate data by simulating a Gaussian diffusion process, which includes both the forward and reverse process. The overall diffusion process is illustrated in Fig. 1. The forward process iteratively adds noise to the image $x_0 \sim q(x)$ through T times of accumulation to obtain $x_0, x_1, \ldots, x_T$. As the number of iterations approaches infinity, the generated sequence of samples tends towards a Gaussian distribution, and it can also be regarded as a Markov process:

$$q\left(x_t|x_{t-1}\right) = N\left(x_t; \sqrt{1-\beta_t}x_{t-1}, \beta_t I\right) \qquad (1)$$

$$q\left(x_{1:T}|x_t\right) = \prod_{t=1}^{T} q\left(x_t|x_{t-1}\right) \qquad (2)$$

where $\{\beta_t \in (0,1)\}_{t=1}^{T}$ is a variance schedule. In this process, as $t$ increases, $x_t$ becomes closer to the noise.

By contrast, the reverse process is the denoising inference process of diffusion. We reconstruct the original image $x_0$ from the standard Gaussian distribution $x_T \sim \mathcal{N}(0, I)$. Since the reversed distribution $q(x_{t-1}|x_t)$ is not computable, the diffusion model to predict such a reversed distribution,

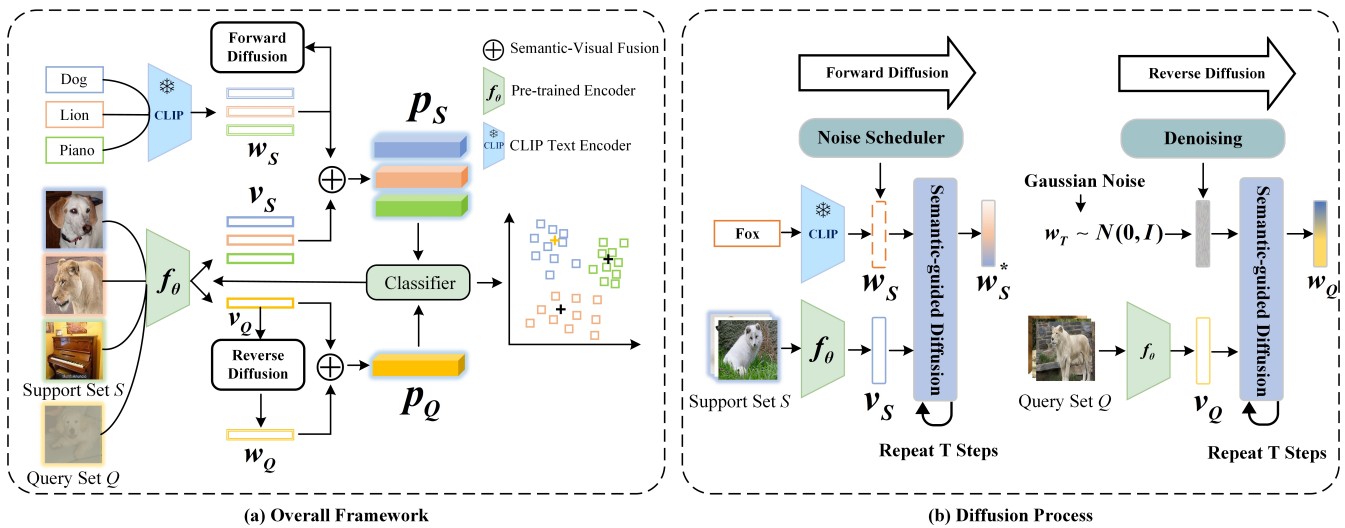

**(a) Overall Framework**      **(b) Diffusion Process**

Fig. 2. The overall structure of the SDPN. The framework includes three stages, where the forward diffusion generates the semantic features based on visual features on the support set. Then the reverse diffusion generates the semantic features of the query set. Afterwards, the fusion strategy combines the visual and semantic features of the sample.

$p_\theta (x_{t-1}|x_t)$, as in (3), where the parameter of the generation process is denoted as $\theta$.

$$p_\theta (X_{0:T}) = p(x_T) \prod_{t=1}^{T} p_\theta (x_{t-1}|x_t) \qquad (3)$$

where

$$p_\theta (x_{t-1}|x_t) = N\left(x_{t-1}; \mu_\theta (x_t, t), \sigma_t^2 I\right) \qquad (4)$$

$$\mu_\theta (x_t, t) = \frac{1}{\sqrt{\alpha_t}} \left( x_t - \frac{1-\alpha_t}{\sqrt{1-\bar{\alpha}_t}} \varepsilon_\theta (x_t, t) \right) \qquad (5)$$

where $\alpha_t = 1 - \beta_t$, and $\bar{\alpha}_t = \prod_{i=1}^{T} \alpha_i$, $N$ is a Gaussian distribution, $\mu$ is the mean and $\sigma$ is the variance. The primary objective of the training process is to minimize the mean square error (MSE) between the predicted $\varepsilon_\theta$ and the actual noise $\varepsilon$ [18], as shown in (6):

$$L_\theta = E_{t,x_0,\varepsilon} \left( \left\| \varepsilon - \varepsilon_\theta \left( \sqrt{\bar{\alpha}_t} x_0 + \sqrt{1-\bar{\alpha}_t} \varepsilon, t \right) \right\|^2 \right) \qquad (6)$$

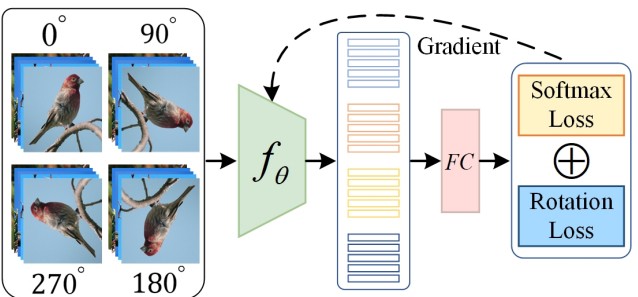

Fig. 3. The pre-training process.

Then, we can sample from Gaussian noise $x_t$ and gradually denoise it to generate the image $x_0$:

$$x_{t-1} = \frac{1}{\sqrt{\alpha_t}} \left( x_t - \frac{1-\alpha_t}{\sqrt{1-\bar{\alpha}_t}} \varepsilon_\theta (x_t, t) \right) + \sigma_t \varepsilon \qquad (7)$$

Choi et al. [18] added conditions into the dependent variable of Denoising Diffusion Probabilistic Models (DDPM) [16], resulting in the generation of high-quality and great-diversity images. Moreover, the diffusion model's sampling strategy has allowed it to be utilized in various downstream tasks, including translation from image to image and generation from text to image, achieving overwhelming performance. Therefore, inspired by using additional information to generate class-specific images, we propose a Semantic-guided Diffusion Prototypical Network (SDPN), which comprehensively analyzes image and label information in FSC tasks to generate class-specific prototypes.

## III. METHOD

This section provides a brief overview of our proposed Semantic-guided Diffusion Prototypical Network, and then describe each stage in details.

### A. Method Overview

For few-shot classification, we typically partition the base dataset into two sets: the base classes, denoted as $C_b$, and the novel classes, denoted as $C_n$, where $C_b \cap C_n = \varnothing$. For an N-way K-shot FSC task, there are two sets of samples used: i) the support-set, which has N classes and K samples per class, and ii) the query-set, which has Q unlabeled samples per class. The i-th N-way K-shot problem be denoted as $T_i = \{S, Q\}$, which includes the support set S and the query set Q. Meanwhile, the objective of FSC is to accurately categorize the query set by utilizing limited samples. The proposed Semantic-guided

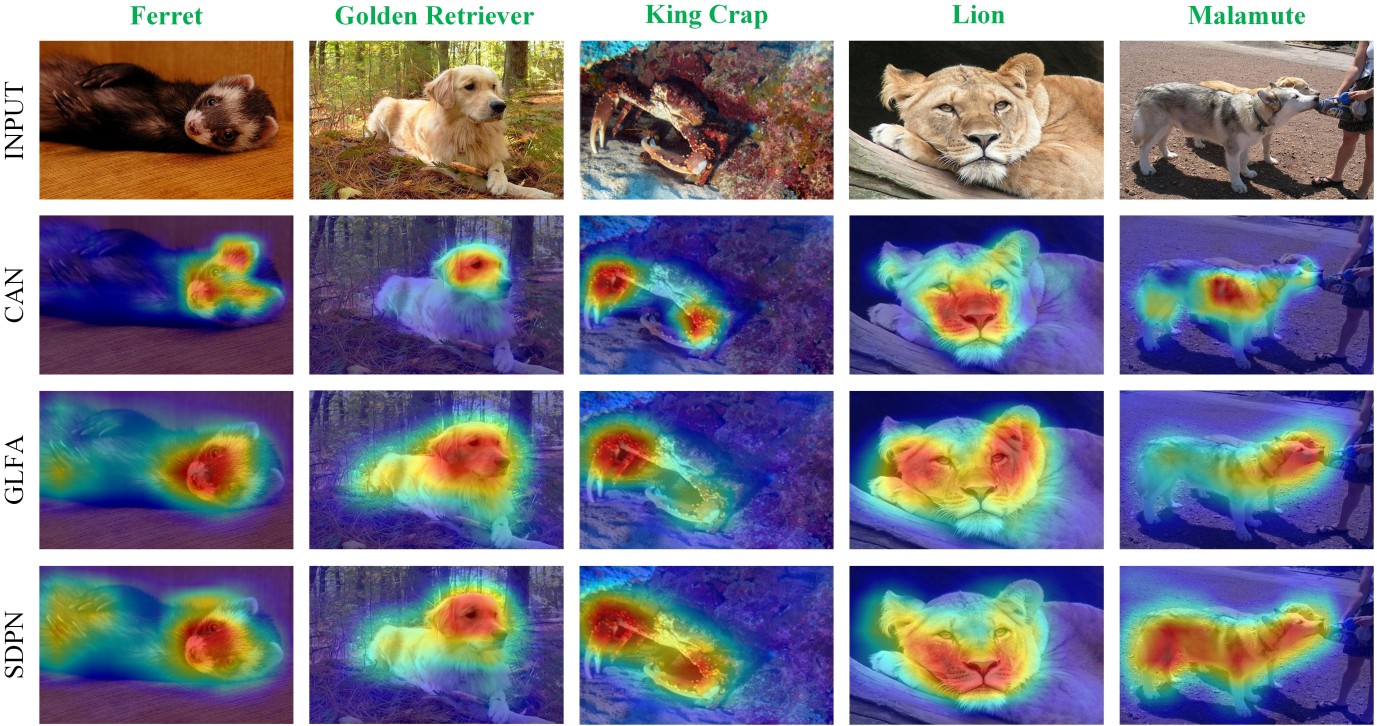

Fig. 4. The attention maps of CAN [4], GLFA [19] and our proposed SDPN based on CAM [20]. We randomly select five images from the miniImageNet dataset [12].

Diffusion Prototypical Network (SDPN) includes three stages, which are pre-training, forward diffusion and reverse diffusion, as shown in Fig. 2. Specifically, we train the feature extractor using a set of base classes to predict rotated angles of an image to improve feature representation during the pre-training stage. During the forward diffusion stage, the SDPN is trained with the support set, which generates diffused semantic features by using the ground-truth (GT) label. During the reverse diffusion stage, we add random noise into the reverse diffusion process to generate the query set's semantic features. Afterwards, we fuse the samples' semantic and visual features to generate more representative prototypes, enhancing the classification accuracy.

### B. Pre-training

We introduce an auxiliary rotation loss to train the features extractor $f_\theta$ that encodes the visual information for FSC tasks. More precisely, we rotate the image using four angles, i.e., $0^\circ, 90^\circ, 180^\circ$ and $270^\circ$, and $f_\theta$ predicts the rotation angles of these rotated images via the auxiliary rotation loss, as shown in Fig. 3. The auxiliary rotation loss [21] is defined as:

$$L_R = min_\theta \frac{1}{GF} \sum_{i=1}^{G} \sum_{j=1}^{F} l\left(f_\theta\left(x_{i,\,j}\right), y\right) \quad (8)$$

where $G$ represents the number of input samples and $F = 4$ denotes the four rotation angles. The $x_{i,\,j}$ indicates the i-th sample with the rotation angle $j$, and $l\left(\cdot\right)$ stands for the cross-entropy loss.

### C. Semantic-guided Diffusion

Due to the particularity of the FSC task, we need to obtain the semantic features of the sample and improve the accuracy and reliability of the prototype by fusing visual and semantic features. However, the labels of the query set are not accessible, so we propose a semantic-guided diffusion process in forward diffusion and reverse diffusion stage. During forward diffusion, for the support set, we use the noise scheduler to add noise to semantic feature $w_S$ of to generate the noisy feature $w_S^*$, and compute the prototype $v_S$ via averaging the visual features. Additionally, we input the noisy feature $w_S^*$ and the prototype $v_S$ into the semantic-guided diffusion module, which generates the diffused semantic vector $\bar{w}_S$. The goal is to generate the denoised semantic features of the support set by minimizing the simplified variational lower bound $L_D$:

$$w_S^* = \sqrt{\bar{\alpha}_t} w_S + \sqrt{1 - \bar{\alpha}_t}\varepsilon \quad (9)$$

$$\bar{w}_S = t_\theta\left(w_S^*, v_S\right) \quad (10)$$

$$L_D = \left| w_S - t_\theta\left(\sqrt{\bar{\alpha}_t} w_S + \sqrt{1 - \bar{\alpha}_t}\varepsilon, v_S\right) \right|^2 \quad (11)$$

Here, $t_\theta\left(\cdot, \cdot\right)$ is achieved by the transformer model [24], which is utilized to obtain the diffused semantic vector $\bar{w}_S$.

In reverse diffusion, for a new FSC task, we input both $v_Q$ and random noise $\varepsilon$ to the learned semantic-guided diffusion process, which generates the diffused semantic feature $w_Q^{T-1}$ of the query set.

| Method | Type | Backbone | MiniImageNet | | TiredImagenet | |
|---|---|---|---|---|---|---|
| | | | *5-way 1-shot* | *5-way 5-shot* | *5-way 1-shot* | *5-way 5-shot* |
| MetaOpt [6] | Optimization | ResNet-12 | 65.64 ± 0.20 | 78.72 ± 0.15 | 68.50 ± 0.92 | 80.60 ± 0.71 |
| CAN [4] | Metric | ResNet-12 | 63.85 ± 0.48 | 79.44 ± 0.34 | 69.89 ± 0.51 | 84.23 ± 0.37 |
| Meta-Baseline [8] | Metric | ResNet-12 | 63.17 ± 0.23 | 79.26 ± 0.17 | 68.62 ± 0.27 | 83.29 ± 0.18 |
| FEAT [3] | Metric | ResNet-12 | 66.78 ± 0.20 | 82.05 ± 0.14 | 70.80 ± 0.23 | 84.79 ± 0.16 |
| AM3-PNet [22] | Metric | ResNet-12 | 65.21 ± 0.30 | 75.20 ± 0.27 | 67.23 ± 0.34 | 78.95 ± 0.22 |
| SAPENet [23] | Metric | ResNet-12 | 66.41 ± 0.20 | 82.76 ± 0.14 | 68.63 ± 0.23 | 84.30 ± 0.16 |
| GLFA [19] | Metric | ResNet-12 | 67.25 ± 0.36 | 82.80 ± 0.30 | 72.25 ± 0.40 | **86.37 ± 0.27** |
| **SDPN (Ours)** | Metric | ResNet-12 | **67.32 ± 0.22** | **83.52 ± 0.21** | **72.75 ± 0.20** | 85.15 ± 0.32 |

After T iterations, we obtain the final semantic feature $w_Q = t_\theta (v_Q, \varepsilon)$ of the query set, where $\varepsilon$ is standard Gaussian variable given by $\varepsilon \sim N(0, I)$.

### D. Prototype Generation

First, we use pre-trained feature extractor to obtain visual features. Then, we compute visual prototypes $v_n$ by averaging the support set's visual features for each class.

$$v_n = \frac{1}{K_S} \sum_{(x_j,y_j) \in S_n} f_\theta (a_j), n = 1, \ldots, N \qquad (12)$$

where $S_n$ represents a subset of $S$ that includes samples from n different classes. Then, the semantic features $w_n$ are denoted as label embedding extracted from CLIP [11]. Finally, we construct the final prototypes by combining $v_n$ with $w_n$ based on the fusion strategy, which is denoted by:

$$p_n = \lambda v_n + (1 - \lambda) w_n, n = 1, \ldots, N \qquad (13)$$

To infer the labels of the query set, we measure the Euclidean distance between $q_i$ and $p_n$.

$$d(q_i, p_n) = \|q_i - v_n\|_2, i = 1, \ldots, NK_Q \qquad (14)$$

$$p(\hat{y}_i = n|q_i) = \frac{exp(-d(q_i, p_n))}{\sum_{n=1}^{N} exp(-d(q_i, p_n))}, n = 1, \ldots, N \qquad (15)$$

where $q_i$ denotes the i-th query prototype, $v_n$ represents visual prototype of the n-th support set and $\hat{y}_i$ represents the predicted label of the i-th query sample.

Accordingly, the overall loss function of the SDPN uses a weighted loss strategy:

$$L(\varphi) = \sum_{i=1}^{Q} [-\log p(y^{q_i}|x^{q_i}, \varphi) + \beta L_D] \qquad (16)$$

where $Q$ represents the number of samples for each query set, $\beta$ represents the hyperparameter of the feature extractor.

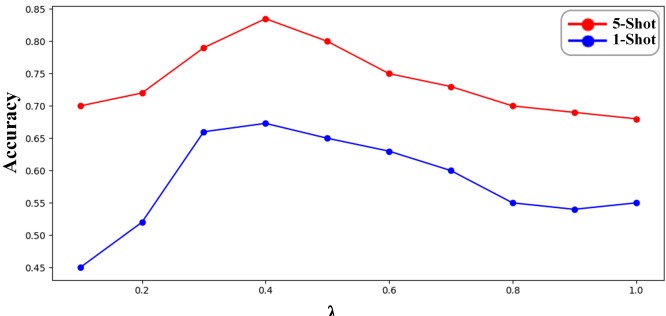

Fig. 5. Accuracy for different values of $\lambda$.

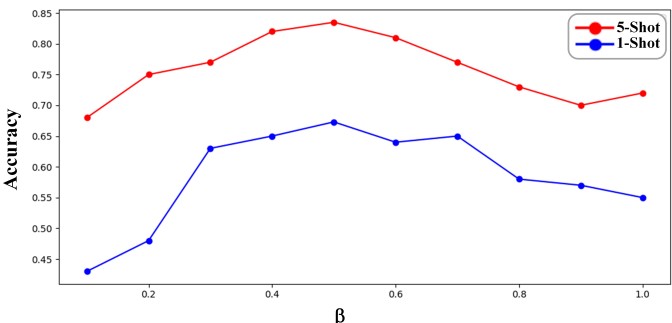

Fig. 6. Accuracy for different values of $\beta$.

## IV. EXPERIMENT

We compared the SDPN to related state-of-the-art methods to evaluate its performance on two publicly available datasets, namely miniImagenet [12] and tieredImagenet [13]. Furthermore, the ablation studies are performed on miniImagenet [12] to assess the performance of each stage.

### A. Datasets and Evaluation Metric

The miniImagenet [12] is a part of the ImageNet dataset [25]. It includes 100 categories, and each category contains a

total of 600 images. It was divided into 64, 16, 20 classes for training, validation and test.

The tieredImagenet [13] is a part from the ImageNet dataset [25], consisting of 608 categories, and each category contains a total of 1200 images. It was divided into 20, 6, 8 classes for training, validation and test.

### B. Implementation Details

We utilize the ResNet12 as the feature extractor which is consistent with the majority of previous studies [3], [4], [6]. The features are obtained by taking an average of the outputs from the final residual block. Our semantic-guided diffusion network is based on the transformer [24]. For miniImageNet [12], we perform training with 100 epochs, setting the batch size to 128. Moreover, the learning rate decays every 25 epochs. For tieredImageNet [13], we used batch size of 128 and ran 100 epochs. Meanwhile, the learning rate decays at epoch 20 and 60. The studies were conducted using a PyTorch 3.9 framework on an NVIDIA GeForce RTX 3080 GPU.

### TABLE II
ABLATION STUDY OF DIFFERENT MODULES

| Module | | | 1-shot | 5-shot |
|---|---|---|---|---|
| Pre | Rotate | SGN | | |
| × | × | × | 49.42 ± 0.78 | 68.20 ± 0.66 |
| ✓ | × | × | 63.17 ± 0.23 | 79.26 ± 0.17 |
| ✓ | ✓ | × | 64.27 ± 0.13 | 81.96 ± 0.32 |
| ✓ | ✓ | ✓ | **67.32 ± 0.22** | **83.52 ± 0.21** |

### TABLE III
ABLATION STUDY OF SEMANTIC EMBEDDING

| Semantic Extractor | 1-shot | 5-shot |
|---|---|---|
| Word2Vec [26] | 66.55 ± 0.36 | 81.90 ± 0.55 |
| CLIP [11] | **67.32 ± 0.22** | **83.52 ± 0.21** |

### C. Results

We perform numerous experiments on miniImageNet [12] and tieredImageNet [13]. Table I shows the results when comparing our proposed SDPN to the existing FSC approaches. Here, all methods employ the ResNet12 architecture as the backbone. Overall, the SDPN achieves competitive performance to existing methods. In the 5-way 1-shot and 5-way 5-shot settings, our SDPN surpasses GLFA [19] by 0.07% and 0.72% respectively on miniImageNet [12]. The SDPN performs better than the metric-based method SAPENet [23] by 4.12% and 0.85% respectively on tieredImageNet [13]. Furthermore, the SDPN performs better than the semantic-based method AM3-PNet [22] by 2.11% and 8.32% respectively on

miniImageNet [12]. In contrast to AM3-PNet [22], which only extracts word embeddings for category labels in the support set, our method utilizes the semantic information of the support set and query set. This enables a better understanding of the data and results in enhanced classification performance.

### D. Ablation Study

To explore the performance of overall framework, we perform comprehensive experiments using the miniImageNet [12] dataset under the 5-way setting.

*a) The effects of different modules:* We analyze the effects of various combinations of three modules. The results show that each of the three modules designed in SDPN are effective, as demonstrated in Table II.

*b) The effects of Semantic Embedding:* In our experiments, we utilize CLIP [11] to extract semantic features. We train our model with Word2Vec [26] and CLIP as the Semantic Embedding, and the corresponding results are presented in Table III. Here, the CLIP model is trained using a dataset of 400 million pairs consisting of images and their corresponding text titles. In addition, Word2Vec is a model for efficient training of word vectors.

*c) The influence of the hyperparameters:* Fig. 5 describes the results obtained by varying the hyperparameter $\lambda$, which represents the hyperparameter of the fusion strategy when combining the visual prototype and semantic prototype. The analysis indicates that $\lambda = 0.4$ can achieve best performance in the 1-shot and 5-shot settings for miniImageNet. Fig. 6 presents the results of various values on hyperparameter $\beta$, which represents the loss weight of the diffusion process. The analysis indicates that $\beta = 0.45$ can achieve best performance in the 1-shot and 5-shot settings for miniImageNet. The red line and blue line represent distinct settings of 5-shot and 1-shot, respectively.

### E. Visualization

To prove the performance of SDPN, we employ CAM [20] to generate attention maps for representative classes. By examining the attention maps generated by CAN [4], GLFA [19], and SDPN, as depicted in Fig. 4. In particular, it indicates that SDPN can identify the regions that are more significant and relevant. For the example image from the malamute class, the SDPN can capture more meaningful information than both CAN [4] and GLFA [19].

### F. Conclusion

In this work, we introduce a Semantic-guided Diffusion Prototypical Network for FSC tasks, which consists of three stages, i.e., pre-training, forward diffusion and reverse diffusion. The pre-training stage improves the reliability of the feature extractor via self-supervised learning by using auxiliary rotation loss. The forward diffusion stage learns to transform the visual features into semantic features of the support set. The reverse diffusion stage generates the query set's semantic features by conditioning random noise on a limited sample of a given new task, which generates representative support prototypes and query prototypes. Numerous experiments conducted

on two benchmark datasets achieve competitive performance compared to related methods in FSC tasks.

Although our method obtains satisfactory performance in few-shot learning, it faces a few limitations, such as the computational complexity of hyperparameter optimization. We need to optimize different hyperparameters for different datasets. Furthermore, noise sensitivity and higher inter-class similarity can limit performance in few-shot classification. In our future work, we will concentrate on executing adaptive hyperparameter optimization and improving the robustness to noise. In addition, we aim to improve our method to promote inter-class distinction and integrate prompt learning to enhance the effectiveness of our model.

## ACKNOWLEDGMENT

This work was supported by National Natural Science Foundation of China under Grant 62373280.

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
