# OpenReview forum: "Semantic-guided Diffusion Prototypical Network for Few-shot Classification"
_IEEE.org/ICIST/2024/Conference — IEEE ICIST 2024 Conference Submission_

### Official Review · Reviewer_5Jac · 2024-08-25
**Accept, some mistakes should be revised**

**Rating:** 7
**Confidence:** 3

**Review:**

1. The overall structural description should be added to the Introduction.
2. The title of Section III is improper and should be modified.
3. Fig.4 and Fig.5 contain a lot of information with insufficient descriptions in the paper.

---

### Official Review · Reviewer_DYUK · 2024-08-30
**Accept**

**Rating:** 10
**Confidence:** 5

**Review:**

1.	It is suggested that the paper provide a more detailed comparison with existing methods that use semantic information, such as those leveraging CLIP or other multi-modal approaches.
2.	It would be useful to include more discussions on potential limitations or challenges in the proposed approach, such as computational complexity or scalability issues.
3.	The paper should include more comprehensive analyses, such as ablation studies to show the impact of each component (e.g. semantic-guided diffusion, visual-semantic fusion) on the overall performance.

---

### Official Review · Reviewer_tpB2 · 2024-09-03
**Semantic-guided Diffusion Prototypical Network for Few-shot Classification**

**Rating:** 7
**Confidence:** 4

**Review:**

1. The description of the proposed Semantic-guided Diffusion Prototypical Network is detailed and comprehensive. To further enhance accessibility, the authors could include additional figures and diagrams to visually illustrate the pre-training, forward diffusion, and reverse diffusion stages, as well as the prototype generation process. This would make the technical details and mechanisms behind SDPN more intuitive and easy to follow.

2.The introduction covers extensive background on few-shot learning. To enhance clarity, the authors could concisely summarize the key limitations of existing approaches and explicitly state how SDPN addresses these limitations, thereby highlighting its unique contribution and positioning.

---

### Decision · Program_Chairs · 2024-09-06

Accept (Oral)